# Spiro-Oxindole Skeleton Compounds Are Efficient Inhibitors for Indoleamine 2,3-Dioxygenase 1: An Attractive Target for Tumor Immunotherapy

**DOI:** 10.3390/ijms23094668

**Published:** 2022-04-23

**Authors:** Daojing Yan, Jiakun Xu, Xiang Wang, Jiaxing Zhang, Gang Zhao, Yingwu Lin, Xiangshi Tan

**Affiliations:** 1Department of Chemistry, Fudan University, 2005 Songhu Road, Shanghai 200433, China; djyan16@fudan.edu.cn (D.Y.); 17110220109@fudan.edu.cn (X.W.); 2Key Laboratory of Sustainable Development of Polar Fisheries, Ministry of Agriculture and Rural Affairs, Yellow Sea Fisheries Research Institute, Chinese Academy of Fishery Sciences, Laboratory for Marine Drugs and Byproducts of Pilot National Laboratory for Marine Science and Technology, Qingdao 266071, China; xujk@ysfri.ac.cn; 3Key Laboratory of Synthetic Chemistry of Natural Substances, Shanghai Institute of Organic Chemistry, Chinese Academy of Sciences, 345 Lingling Road, Shanghai 200032, China; zhangjiaxing@chemchina.com (J.Z.); zhaog@mail.sioc.ac.cn (G.Z.); 4School of Chemistry and Chemical Engineering, University of South China, Hengyang 421001, China

**Keywords:** indoleamine 2,3-dioxygenase 1, metalloenzyme, hemeprotein, inhibitor, molecular docking

## Abstract

Indoleamine 2,3-dioxygenase 1 (IDO1) is an attractive heme enzyme for its significant function in cancer immunotherapy. Potent IDO1 inhibitors have been discovered for decades, whereas no clinical drugs are used for cancer treatment up to now. With the goal of developing medically valuable IDO inhibitors, we performed a systematic study of SAR405838 analogs with a spiro-oxindole skeleton in this study. Based on the expression and purification of human IDO1, the inhibitory activity of spiro-oxindole skeleton compounds to IDO1 was evaluated by IC_50_ and *K*_i_ values. The results demonstrated that inhibitor **3** exhibited the highest IDO1 inhibitory activity with IC_50_ at 7.9 μM among all inhibitors, which is ~six-fold of the positive control (4−PI). Moreover, inhibitor **3** was found to have the most effective inhibition of IDO1 in MCF-7 cancer cells without toxic effects. Molecular docking analysis revealed that the hydrophobic interaction stabilized the binding of inhibitor **3** to the IDO1 active site and made an explanation for the uncompetitive mode of inhibitors. Therefore, this study provides valuable insights into the screen of more potent IDO1 inhibitors for cancer immunotherapy.

## 1. Introduction

Proteins have many applications, for example, vaccines and antibodies are hotspots of research for disease treatments in recent years [1,2]. Meanwhile, proteins can also act as pharmacological targets for the treatment of diseases, such as cancer [3,4], inflammation, and autoimmune disorders [5,6]. According to statistics, more than 5700 clinical trials of ligand-targeting drugs have been registered since 1992 [6], which highlights the significant application prospects of protein targets.

Metalloproteins such as heme enzymes play vital roles in biological systems and others [7,8,9,10,11]. Indoleamine 2,3-dioxygenase (IDO1) is a cytosolic heme protein [12] and is one of the key metabolic enzymes, which makes tryptophan (Trp) metabolism along the kynurenine pathway (KP) [13,14]. It has been observed that IDO1 is an attractive target for the pathogenesis of neuroinflammatory and neurodegenerative disorders, such as Alzheimer′s disease, depression, cataracts, and HIV encephalitis [15]. More importantly, IDO1 has been shown to play an important role in the process of immune evasion by tumor. Generally, the important function of IDO1 in tumor immunology is accepted for two reasons. For one reason, the high expression of IDO1 in tumor cells caused depletion of local Trp, which severely affected the proliferation of T lymphocytes [16]. For the other reason, kynurenine (Kyn), the initial metabolite of Trp, is an endogenous immunologically active ligand for the aryl hydrocarbon receptor (AhR). The activation of AhR accelerates the conversion of effector T lymphocytes into regulatory T cells, and then may upregulate IDO1 expression in DC cells, expanding immunoregulatory effects and blocking anti-tumor immune response of the body further [17,18]. Therefore, IDO1 has obtained great attention as a therapeutic target with great potential in cancer immunotherapy in past decades. However, there is no targeting IDO1 drug applied to tumor treatment at present [19]. Thus, intensive efforts should be made to investigate the efficient IDO1 inhibitors for cancer treatment.

In this study, we investigated a series of spiro-oxindole skeleton compounds for testing their IDO1 inhibitory activities, and the chemical structure of the compounds is exhibited in Figure 1. Enormous evidence suggests that spiro-oxindole core structure confers valuable bioactivity and pharmaceutical properties [20,21,22]. Meanwhile, SAR405838 is an example of spiro-oxindole skeleton compound reported as a powerful and selective inhibitor of the MDM2 target for cancer therapy [23]. The clinical trials of this candidate for anti-tumor efficacy are in phase I [24]. Moreover, the core structure of compounds investigated in our work has a structural similarity to tryptophan analog, which is an important route for IDO1 inhibitor investigation [25]. Furthermore, indoximod (1-Methyl-D-Tryptophan, 1-D-MT), a tryptophan analog, is a successful case for cancer treatment in clinical II stage targeting IDO1 [26]. According to the latest data, indoximod in combination with pembrolizumab showed antitumor efficacy worth further evaluation in patients with advanced melanoma [27]. For these reasons, we believe that spiro-oxindole skeleton compounds may have a remarkable inhibitory activity on IDO1.

## 2. Results and Discussion

### 2.1. UV–Vis Spectra of IDO1 and Reversibility Analysis of Inhibitors

As shown in Figure 2A, the molecular weight of IDO1 was identified by SDS-PAGE gel with a mass of around 45 kD. UV–vis spectra displayed the characteristic absorptions of IDO1 in different forms (ferric, met-form: 404 nm, 500 nm, and 635 nm; ferrous, deoxy form: 429 nm and 559 nm; Figure 2B), which were in accordance with those reported [28]. Since IDO1 is a heme-containing enzyme, and the iron porphyrin in the active site of IDO1 is highly sensitive to the binding of substrate/ligand, which could be reflected in UV–vis spectrum [25], therefore, this method can be used for the selection of the IDO1 inhibitor preliminarily.

In our experiments, four spiro-oxindole skeleton compounds were obtained from the lab of Professor Zhao [29,30]. The synthetic method and identification of these compounds are displayed in the Appendix A. From the chemical structures, we can clearly see that compounds **1**, **2**, and **3** had structural differences at 6-position, and compound **4** was a trans structure with a difference at 2’-position.

The experiment of evaluating the binding of compounds 1 to 4 with IDO1 was performed by UV–vis spectrometer. Oxidized ferric and reduced ferrous states of IDO1 in mixing with compounds 1 to 4 are presented in Figure 3. Compared with the pure protein of oxidized form, the highest absorption of IDO1 compounds still appeared at 404 nm without other changes in UV–vis spectra except for the difference in absorption intensities (Figure 3A). However, there was a difference between the pure protein of reduced form and IDO1 compounds in UV–vis spectra. Figure 3B showed that the Soret band of IDO1 compounds exhibited a blue shift from 429 nm to 424 nm, and the Q band shifted from 560 to 556 nm. Excitingly, a unique peak appeared in each spectrum at 524 nm. These spectral changes in Figure 3B demonstrated that four compounds could interact with the IDO1 active center. These findings laid an important foundation for the study of spiro-oxindole skeleton compounds as IDO1 inhibitors.

Based on the results of UV–vis spectra, the determination of inhibitory types was carried out according to the reference [31]. The inhibitor was defined as reversible if the fitted lines were intersected in the plot (Figure 4A). The data analysis showed that four inhibitors were categorized as reversible inhibitors, and inhibitor **3** was selected as a typical sample here (Figure 4B).

### 2.2. Enzyme-Based IDO1 Inhibitory Assay to Determine the IC_50_ and K_i_ Values

In order to evaluate the enzymatic IDO1 inhibitory activity of spiro-oxindole skeleton compounds, 4−Phenylimidazole (4−PI) was selected as a positive control. IC_50_ values for inhibitors were examined at the same substrate concentration (100 μM Trp). The results was presented in Figure 5. 4−PI showed IC_50_ value as 49.2 μM, which was consistent with the reported result (IC_50_ = 48 μM) [32]. From the determination of IC_50_ value, we can see that inhibitors **1**, **2,** and **3** had better IDO1 inhibitory activities around 30 μM than 4−PI. However, there was a slight difference between inhibitor **1** (25.2 μM) and **2** (29.4 μM) which may be caused by the methyl group in 6-position. For the methyl group, electron donors could weakly impact the interaction between the IDO1 and inhibitor. However, the significant effect of the electron donor in 6-position was observed in inhibitor **3**. The lowest IC_50_ value of **3** (7.9 μM) was owing to the electron-donating property of the methoxyl group, which is stronger than the methyl group [33]. This could cause a dramatic interaction between the inhibitor and residues in the IDO1 active center. This small modification of the substituent group made inhibitor **3** become the optimal IDO1 inhibitor which increased the inhibitory activity by 3 times. Interestingly, inhibitor **4** (64.6 μM) was an anti-conformation with the lowest inhibitory efficacy of the four inhibitors. The spiro-oxindole skeleton compounds with anti-conformation were unlikely to match the structural cavity of IDO1.

Most drugs function through enzyme inhibition interactions with their targets with simple, reversible binding mechanisms [34]. The mode of reversible inhibition is further divided into three types of inhibitors, including competitive, non-competitive, and uncompetitive. Specifically, a competitive inhibitor binds exclusively to the free enzyme. A noncompetitive inhibitor is one that displays a binding affinity for both the free enzyme and the enzyme–substrate (ES) complex or subsequent species. The definition of uncompetitive inhibitor is that it only recognizes and interacts with the ES complex and subsequent downstream catalytic species with no binding to free enzymes [34,35]. In our work, the determination of inhibitor modes was based on reversible results of four spiro[pyrrolidin-3,3′-oxindoles] compounds (Figure 4B). Furthermore, the author Cornish-Bowden reported a simple graphical method for determining the inhibition constants (*K*_i_) and inhibitor modes [36], which is a common method in the discovery of inhibitors [37,38].

*K*_i_ value is a key parameter of inhibitors in drug research, which is the dynamic process between drug and target, representing the dissociation constant of the enzyme–inhibitor complex, or the reciprocal of the binding affinity of the inhibitor to the enzyme [39]. According to the graphical method, *K*_i_ values of the four inhibitors in our work were obtained by intersection of the three fitted lines in plot, which was the absolute value of the *X*-axis value of the intersection. Inhibitor modes were determined by the position of the intersections in the plot. Therefore, our results clearly demonstrated that four inhibitors were uncompetitive binding modes, while the positive control (4−PI) was a noncompetitive inhibitor consistent with the reported result (Figure 6) [40]. It revealed that spiro[pyrrolidin-3,3′-oxindoles] compounds recognize and interact with the IDO1-Trp complex to exert their inhibitory efficacy. In addition, the *K*_i_ values of inhibitors are summarized in Table 1, which showed that inhibitor **1** had the best affinity with IDO1 (*K*_i_ = 42.3 μM) compared with others. However, the *K*_i_ value of inhibitor **3** (63.6 μM) was ~1.5 times higher than **1**. By comparison, inhibitor **3** was considered the most potent IDO1 inhibitor for the acceptable *K*_i_ value and best inhibitory activity in terms of the IC_50_ value.

### 2.3. Cellular IDO1 Inhibitory Activity Assay

To explore the IDO1 inhibitory activity of these spiro-oxindole skeleton compounds, four inhibitors were tested in four cancer cells, in triplicate. Assays were performed with the concentration of inhibitors at 100 μM (Table 2). The results of the analysis clearly showed that these inhibitors had higher inhibition of IDO1 expressed in HeLa and MCF-7 cells than those in CT26 and 4T1 cells. Furthermore, inhibitor **3** exhibited the most potent inhibitory activity in four cell lines and was even more remarkable in MCF-7 cells, while four inhibitors had no selective inhibitory effects in HeLa cells (Figure 7A,B).

To ensure that there were no false-positive results, the cell viability was measured by L02 cells, normal human hepatocytes. The CCK8 kit was used for the detection of cell viability. When four inhibitors were set at the highest concentration (200 μM) in the experiment, the cell viability could reach over 85% (Figure 7C). This result demonstrated that four spiro-oxindole skeleton compounds did not indicate obvious toxicity.

### 2.4. Molecular Docking Results

The Autodock 4 program was used to perform the semi-flexible docking of four inhibitors with IDO1, respectively. According to the rigid docking results, six residues closest to the active center of IDO1 were selected as flexible residues, namely, Phe163, Leu234, Ser263, Ala264, Phe226, and Thr379. Results of VMD 1.9 analysis showed that four inhibitors were all located in the active cavity of IDO1 (Appendix A and Figure 8B). One of the benzene rings in the three inhibitors **1**, **2**, and **3** was located above the heme of IDO1 directly. The three inhibitors had good overlaps which indicated they had similar inhibitory activity to IDO1 (Appendix A). This coincided with our enzyme-based IDO1 inhibitory results (IC_50_ from 7.9 to 25.2 μM). In particular, the position of inhibitor **4** shifted in the active center of IDO1 (Appendix A), which weakened the interaction with the active center of IDO1 (IC_50_ = 64.6 μM). It was the reason why the inhibitor had the weakest inhibitory activity to IDO1, indicating that the trans conformation was not the dominant conformation of this type of inhibitor.

In addition, we analyzed the docking results of inhibitor **3** with IDO1 using LigPlot+, and the results showed that inhibitor **3** was stabilized at the active center of IDO1 through strong hydrophobic interactions (Figure 8A). Subsequently, VMD 1.9 was used to analyze inhibitor **3** in the active center of IDO1 in detail. The results showed that the benzene ring of inhibitor **3** was parallel to the heme of IDO1 with a distance of 2.74 Å. It indicated that inhibitor **3** had hydrophobic interaction with heme, which explained the reasons for the changes in the UV–vis spectra of the inhibitor–IDO1 complex. Moreover, the inhibitors did not bind to heme directly, which revealed that the inhibitors were in uncompetitive modes [35]. The amino acid residues closest to inhibitor **3** displayed by VMD 1.9 were Ser167, Phe226, Arg231, Thr379, and Ala264, and their closest distances to the inhibitor were all about 3.5 Å (Figure 8C). It was worth noting that there was literature showing that Phe226 and Arg231 were important residues affecting substrate binding [41]. Therefore, the strongly hydrophobic interaction of inhibitor **3** with residues Phe226 and Arg231 may be the key reason for the best inhibitory effect of this inhibitor to IDO1. Moreover, ten of the most favorable docking results are listed in tables (Table 3 and Appendix A); four docking energies including binding energy, intermolecular energy, Van der Waals energies and electrostatic interactions between inhibitors, and IDO1 were included. These results showed that inhibitor **3** had the lowest binding energy, demonstrating that the inhibitor **3**–IDO1 complex was the most stable. All the above results indicate that inhibitor **3** had the strongest inhibitory effect on IDO1.

## 3. Conclusions

Based on the expression and purification of IDO1, the interactions between spiro-oxindole skeleton compounds and IDO1 are evaluated primarily by UV–vis spectrophotometry. These results demonstrate that four compounds interacted with the active pocket of IDO1. Subsequently, four compounds are defined as reversible inhibitors of IDO1. The experiments of enzymatic inhibitory activity clearly show that inhibitor **3** is capable of inhibiting IDO1 intensively. It is ~six times higher than the positive control (4-PI) in IC_50_ value. The graphical method presented by Cornish-Bowden indicated that four spiro[pyrrolidin-3,3′-oxindoles] compounds are uncompetitive inhibitors. Meanwhile, inhibitory measurements in cancer cell lines also confirmed that inhibitor **3** had the best effect on MCF-7 cells and was more selective than other cancer cells without toxic effects. Moreover, to provide a better understanding of why inhibitor **3** inhibited IDO1 effectively, the method of molecular docking is used for a full explanation. The docking results display that those four inhibitors are in the active cavity of IDO1 except that the position of inhibitor **4** shifted to the active center. This indicates that inhibitor **4** (trans conformation) is not the favorable conformation of this type of inhibitor. In addition, docking results reveal that the hydrophobic interaction of inhibitor **3** with residues Phe226 and Arg231 is the key reason for the best inhibitory effect of this inhibitor on IDO1. Moreover, inhibitors do not bind to the heme directly, revealing that the inhibitors are in uncompetitive modes.

In conclusion, we synthesize four compounds with spiro-oxindole skeleton and obtain the target protein (IDO1) successfully by biological methods. Our results prove that spiro-oxindole skeleton compounds have an effective inhibitory activity to IDO1. This reveals that the hydrophobic interactions accelerate the stability of the IDO1-inhibitor complex. However, all inhibitory experiments demonstrate that four inhibitors have weak inhibitory efficacy on IDO1. However, structural modification could be made on the spiro-oxindole skeleton in future work for synthesizing more potent IDO1 inhibitors. Therefore, our study provides useful clues for the exploration of more potent human IDO1 inhibitors.

## 4. Materials and Methods

### 4.1. Materials and Reagents

The GST-IDO1 gene was cloned by Shanghai Generay Biotech Co. Ltd (Shanghai, China). The BL21(DE3) cells were obtained from TransGen Biotech Co. Ltd (Shanghai, China). Ampicillin, isopropyl β-D-thiogalactoside (IPTG), 5-Aminolevulinic acid (5-ALA), phenylmethanesulfonyl fluoride (PMSF), DNase I (from bovine pancreas), reduced glutathione, L-tryptophan, ascorbic acid, methylene blue, catalase, and sodium dithionite were obtained from Sangon Biotech (Shanghai, China) Co. Ltd. DMSO was purchased from Sigma-Aldrich. The CCK8 kit was purchased from Beyotime Biotechnology Co. Ltd.(Shanghai, China). Glutathione agarose beads were purchased from Changzhou Smart-Lifesciences Biotechnology Co. Ltd. (Changzhou, China) PD-10 desalting column was purchased from GE Healthcare Biosciences. HRV 3C Protease was purchased from Sino Biological Inc. (Shanghai, China). Cancer cell lines were purchased from the Cell Bank of Chinese Academy of Sciences. The experimental compounds were provided by Professor Zhao (Shanghai Institute of Organic Sciences, Chinese Academy of Sciences). All chemicals of analytical and reagent grade were obtained from commercial sources and used without further purification.

### 4.2. Expression and Purification of IDO1

The recombinant human IDO1 gene was cloned into pGEX-6P-1-based vector with an N-terminal cleavable GST-tag. Protein was expressed in *Escherichia Coli* BL21 (DE3) cells. The transfected bacteria were grown in sterile LB medium (4 liters) containing 0.1 mg/mL ampicillin at 37 °C. Then, cells were allowed to grow until O.D._600nm_ reached ~0.6–0.8. At that time, 5-ALA (0.5 mM) and IPTG (1 mM) were added in each bottle of medium, and the temperature was decreased to 25 °C. Bacteria were collected as a pellet after 16 h. Cells were then lysed in PBS buffer (pH 7.4) and centrifuged to obtain supernatant at 4 °C. Steps of IDO1 purification from supernatant were as follows. Before supernatant loading, glutathione agarose beads were equilibrated by PBS buffer. Notably, the speed of supernatant on the column needed to be controlled slowly. This was to ensure that GST-IDO1 combined with the column tightly. Afterward, IDO1 with N-terminal GST-tag was digested on the column overnight by HRV 3C protease under 16 °C. Then, IDO1 was eluted by PBS buffer and PD-10 desalting column was applied by exchanging buffer into 50 mM Tris containing 100 mM NaCl. Each procedure was confirmed by SDS-PAGE gel. Moreover, the heme incorporation of purified IDO1 was also tested and the ratio of λ_404/280_ was ∼2.0, which approximated the fully heme-incorporated IDO1 (the ratio was around 2.2) [42]. Finally, protein was concentrated to around 10 mg/mL (2 mL) and stored at −80 °C before use.

### 4.3. UV–Vis Spectra of IDO1 and IDO1–Complex

To determine the spectra of IDO1 in different oxidation states, experiments were performed according to the literature [43]. IDO1 (10 μM, 2 mL, 100 mM phosphate buffer, pH 6.5; ɛ_404 nm (IDO1)_ = 172 mM^−1^cm^−1^ [44]) and the sodium dithionite solution (5 M) was prepared. The reduction form of IDO1 was obtained by adding sodium dithionite.

As mentioned in the above, Ox-IDO1 and Red-IDO1 were prepared in the degassing sealed cuvette. Next, 100 μM inhibitor (pipetting 20 μL from 10 mM stock solution which prepared by buffer containing 1% DMSO) was added in two cuvettes, respectively, before starting the scan of the UV–vis spectrophotometer. Then, samples were incubated at 37 °C for 20 min. The final spectra were recorded by apparatus immediately. It was reflected in the spectra whether the inhibitor bound to the iron in the IDO1 active site [25].

### 4.4. Determination of the Inhibition Types

Different concentrations of IDO1 (1 mL) were prepared with 100 mM phosphate buffer (pH 6.5) containing methylene blue (10 μM), catalase (200 μg/mL), and ascorbic acid (10 mM, pH 6.5). Each sample was incubated at 37 °C for 10 min. An amount of 100 μM L-Trp was added before measuring by UV–vis spectrophotometer at 320 nm. The only difference for the other set of groups was 100 μM inhibitor was added in each sample before incubating. The data were analyzed by Origin 7.5, and plot of reaction rate [V] versus enzyme amount [E] was proposed using linear fit. The determination of inhibitory types was based on the linear trend of the fitted plot [31,45].

### 4.5. Enzyme-Based IDO1 Inhibitory Assay

The IDO1 inhibitory assay was performed according to standard literature protocols with some modifications [45]. The UV–vis method was used for evaluating the inhibitory activity of inhibitors against IDO1. Specifically, IDO1 (1 μM, 1 mL) was prepared by phosphate buffer (100 mM, pH 6.5) containing catalase (200 μg/mL), methylene blue (10 μM), and ascorbic acid (10 mM, pH 6.5). Concentrations of every inhibitor were varied from 1 to 200 μM, and experimental samples were incubated at 37 °C for 10 min. Then, a certain concentration of L-Trp was added before monitoring the product at 320 nm by UV–vis spectrophotometer immediately.

The inhibition percent of each inhibitor was determined by [1 − (*A*/*B*)] × 100%, where *A* is the enzymatic activity of IDO1 with inhibitor added and *B* is the absence of the inhibitor. The determination of IC_50_ values was performed at a L-Trp concentration of 100 μM, and the IC_50_ values of each inhibitor were calculated using nonlinear regression with a dose–response–inhibition equation (four parameters) in GraphPad Prism 7.00. The plot was described by the percent inhibition against Log_10_ concentration of the inhibitor.

The same experimental method as mentioned above was used for the determination of the inhibitory constant (*K*_i_), with three different L-Trp concentrations (50 μM, 100 μM, and 150 μM) to obtain the velocity of reaction. *K*_i_ values of inhibitors were obtained by intersection of the three fitted lines in the plot, which was the absolute value of the *X*-axis value of the intersection. The plot was described by [S]/[V] against inhibitor concentration, where [S] and [V] were the substrate concentration and the reaction velocity, respectively [15]. Furthermore, protein-inhibitor binding modes were determined by the position of the intersections in the plot. Data processing was performed by Origin 7.5.

### 4.6. IDO1 Inhibitory Assay in Cell Lines

IDO1 cellular activity assay was carried out by standard literature protocols [46]. IDO1 was highly induced by IFN-γ in cancer cells and was used as a carrier for IDO1 producing. First of all, cells grew in high glucose DMEM medium containing 10% FBS and antibiotics. On the first day for IDO1 inhibitory assay, the cells were harvested from Petri dishes by trypsin/EDTA (1 mL, 0.25%). Then, 1 × 10^4^ cells were seeded in 96-well plates and cultivated in carbon dioxide incubator at 37 °C containing 5% CO_2_ for 24 h. On day two, each well in the plates were treated with fresh DMEM medium, which contained L-Trp (20 μg/mL), IFN-γ (10 ng/mL), and different concentrations of inhibitor. The plate was incubated for 48 h further. Then, 140 μL of supernatant from each well was transferred into centrifuge tubes correlatively. An amount of 10 μL of 30% trichloroacetic acid (*w*/*v*) was added to each tube aimed to precipitate proteins. Subsequently, sample incubation (65 °C, 30 min) was required to make the N′-formylkynurenine (NFK) change into kynurenine which was the final product. Soon afterwards, the sample was centrifuged at 8000 rpm for 10 min. An amount of 100 μL supernatant from each tube was transferred into a new 96-well plate with an equal amount of 2% pDMAB added. The concentration of kynurenine could be measured because it reacts with pDMAB to form a product with a UV–vis absorption at 492 nm. To estimate cell viability, 1 × 10^4^ normal cells named L02 were seeded in 96-well flat bottom plates with a final volume of 200 μL of DMEM complete growth medium. After being incubated for 24 h at 37 °C, the culture was replaced by a fresh complete growth medium (200 μL) and four inhibitors (200 μM) were added individually. Then, 48 h later, the medium was replaced by PBS (180 μL) mixed with CCK8 solution (20 μL) in each well. Cell viability was evaluated by testing the OD_450nm_. Finally, all data in cell experiments were processed by GraphPad Prism 7.00 to obtain the IDO1 inhibitory percentage of each inhibitor in the cells and the cell viability.

### 4.7. Molecular Docking Study

Docking simulations were undertaken to study the binding pattern of four inhibitors in the IDO1 pocket. The structural models of IDO1 and ligand were generated by the following steps. Water molecules and ligand in the human IDO1 structure (PDB code: **5WMU** [47]) were removed from the active site before docking. Heme, a prosthetic group of IDO1, remained its original state in crystal structure. The molecular structure of inhibitor **3** for docking was generated by the Dundee Prodrg2 server [48]. The docking experiments were performed using Autodock 4 program. Setting of a receptor grid file was generated with grid box size of 60 × 60 × 60 Å. Based on the position of the original ligand in crystal structure, the parameters of the center coordinates were set as X = 114.802, Y = −4.805, and Z = 142.988. The Autodock 4 program can assign force field parameters to the receptor and ligand automatically. Moreover, the force field for molecular docking was AMBER based on non-bonding interaction, and this interaction consisted of van der Waals interactions, hydrogen bonding interactions, and electrostatic interactions. The method of molecular docking was based on the lattice point docking and the size of the lattice point was 0.375 angstroms. During the calculation, the lowest binding energy of the ligand with receptor was searched in all lattice points for docking evaluation. A total of 2000 conformations were set in molecular docking and the semi-empirical method for free energy calculation was used for docked conformation ranking automatically. Finally, ten favorable conformations were then visualized and analyzed by VMD 1.9 and LigPlot^+^ for exploration of the binding interactions between ligand and IDO1.

## Figures and Tables

**Figure 1 ijms-23-04668-f001:**
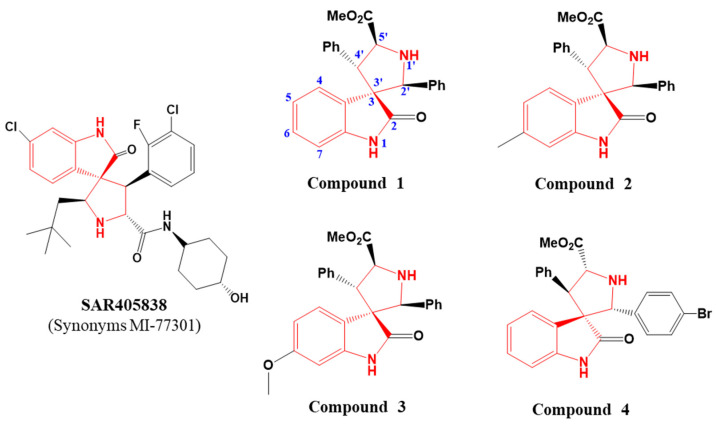
Chemical structures of a series of spiro-oxindole skeleton compounds and an inhibitor targeting MDM2 protein named SAR405838.

**Figure 2 ijms-23-04668-f002:**
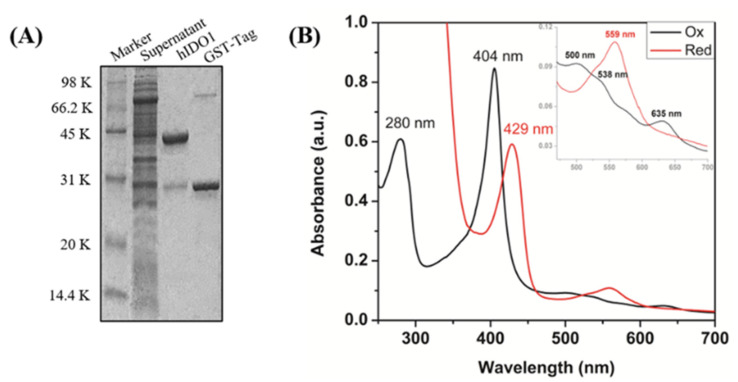
(**A**) Identification of IDO1 in expression and purification. The SDS-PAGE gel shows that GST-IDO1 is soluble after expression and appeared in supernatant with a molecular weight of over 66.2 kD. The strip of target protein is presented at 45 kD after removing the GST-tag. (**B**) UV–vis spectra of IDO1 in ferric and ferrous forms.

**Figure 3 ijms-23-04668-f003:**
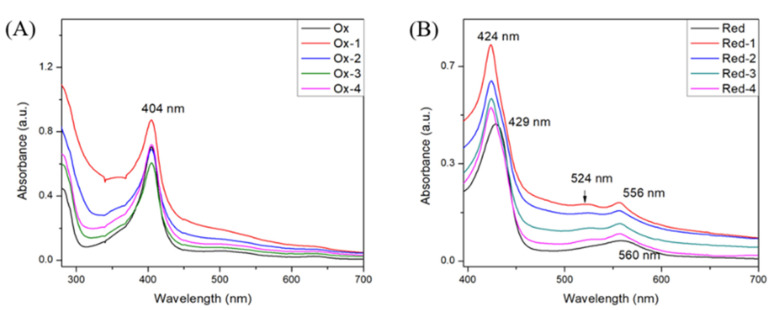
UV–vis spectral analysis of IDO1 and upon the binding of inhibitors. (**A**) UV–vis spectra of IDO1 in the ferric forms with 100 μM concentration of different inhibitors; (**B**) UV–vis spectra of IDO1 in ferrous forms with 100 μM concentration of different inhibitors.

**Figure 4 ijms-23-04668-f004:**
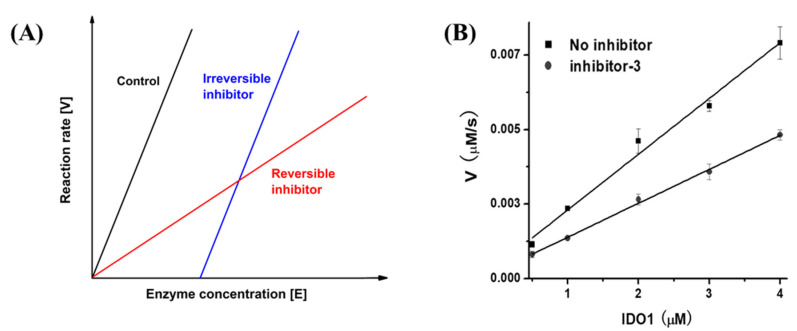
Determination of inhibition types. (**A**) The evaluation standard of inhibition type. The plot is enzymic reaction rate [V] against enzyme concentration [E]. The plot represents that curves with different slopes determine the inhibition type: reversible (in red color) or irreversible (in blue color) [31]. (**B**) An example for inhibition type determination with inhibitor **3**. The slope of inhibitor **3** is consistent with the reversible inhibition type. Three independent assays were performed and the data are presented as mean ± SD with three independent assays.

**Figure 5 ijms-23-04668-f005:**
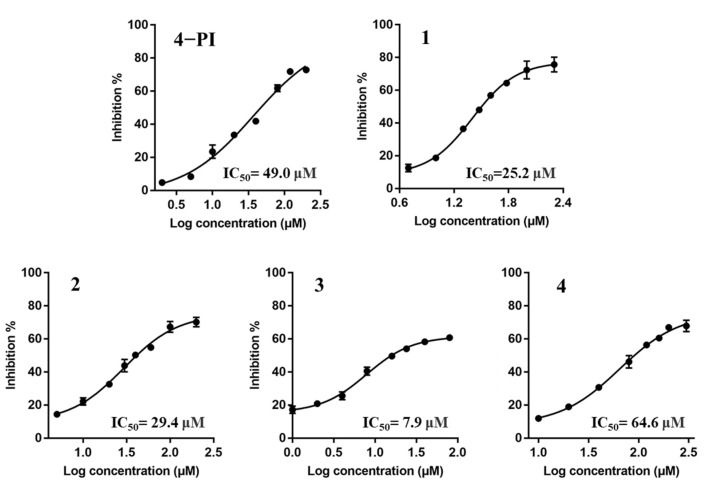
Four inhibitors are tested through IDO1 inhibitory activity assays with 4−PI as a positive control. The IC_50_ values are presented as the mean ± SD with three independent determinations.

**Figure 6 ijms-23-04668-f006:**
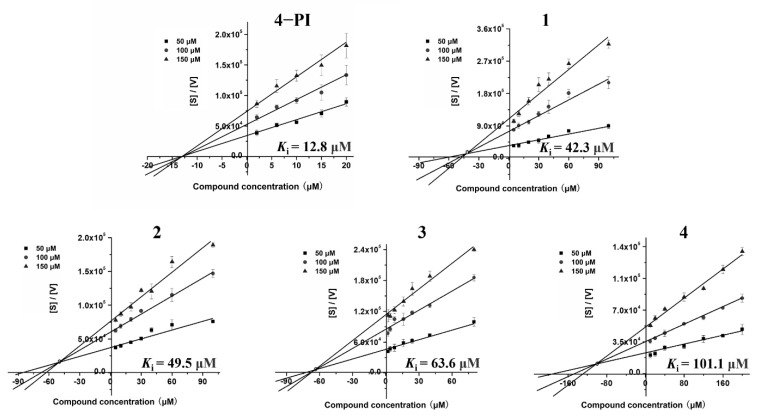
The characterization of spiro-oxindole skeleton compounds as potent IDO1 inhibitors. Kinetic parameters of the four compounds were determined according to the plots of [S]/[V] against compound concentrations [I]. The L-Trp concentrations varied from 50 to 150 μM. The intersection points in plots are used to determine the *K*_i_ values. The *K*_i_ values are presented as the mean ± SD with three independent determinations.

**Figure 7 ijms-23-04668-f007:**
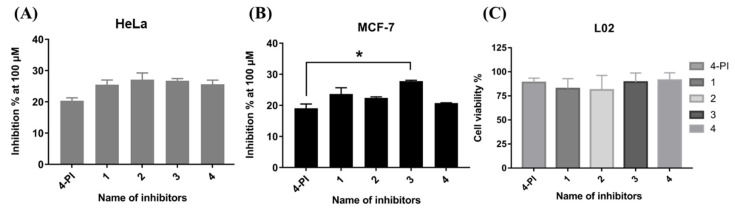
Inhibition percentage of all inhibitors at 100 μM in HeLa (**A**) and MCF-7 cells (**B**); “two-way ANOVA” method is applied for data analysis. The mark “*” means there is a statistical difference between the two groups (*p* < 0.05). (**C**) Cell viability assay for inhibitors using L02. The data are presented as mean ± SD with three independent enzymatic assays.

**Figure 8 ijms-23-04668-f008:**
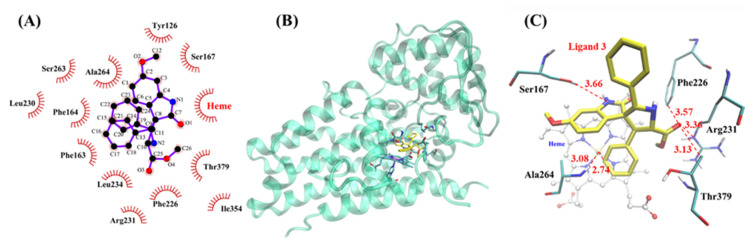
(**A**) Stabilization forces between inhibitor **3** and the active site of IDO1 are defined by LigPlot^+^. For LigPlot^+^ analysis, hydrophobic interactions are illustrated with an “eyelash” symbol. (**B**) The overall structure of the docking results of inhibitor **3** binding to IDO1; inhibitor **3** is shown as stick models in yellow. (**C**) Top view of inhibitor **3**–IDO1 structure. Heme is shown as a ball and stick model in white. Inhibitor **3** and the key hydrophobic residues are shown as stick models in yellow and cyan. Hydrophobic interactions are shown as dotted lines.

**Table 1 ijms-23-04668-t001:** Inhibitory activity of spiro-oxindole skeleton compounds against IDO1 and determination of inhibition types *^a^*.

Compounds	Type of Inhibition	IC_50_ (μM)	*K*_i_ (μM)
**4−PI**	Noncompetitive [40]	49.0 ± 0.7	12.8 ± 2.1
**1**	Uncompetitive	25.2 ± 3.6	42.3 ± 0.8
**2**	Uncompetitive	29.4 ± 1.3	49.5 ± 6.0
**3**	Uncompetitive	7.9 ± 1.7	63.6 ± 3.5
**4**	Uncompetitive	64.6 ± 3.9	101.1 ± 4.4

*^a^* The data are presented as the mean ± SD with three independent enzymatic assays.

**Table 2 ijms-23-04668-t002:** Inhibition (%) at 100 μM inhibitor concentration *^a^*.

Compounds	HeLa Cell	MCF-7 Cell	CT26 Cell	4T1 Cell
**4−PI**	20.3 ± 0.9	19.1 ± 1.4	5.8 ± 0.5	5.7 ± 0.5
**1**	25.5 ± 1.5	23.7 ± 2.0	3.7 ± 0.8	2.6 ± 0.6
**2**	27.1 ± 2.1	22.4 ± 0.4	4.2 ± 0.5	4.2 ± 1.0
**3**	26.7 ± 0.7	27.8 ± 0.3	6.4 ± 0.8	4.3 ± 0.1
**4**	25.6 ± 1.3	20.7 ± 0.1	1.7 ± 0.5	2.8 ± 0.3

*^a^* The data are presented as the mean ± SD with three independent assays.

**Table 3 ijms-23-04668-t003:** Docking results of AutoDock program.

Model	E_binding_ ^a^ (kcal/mol)	E_inter-mol_ ^b^ (kcal/mol)	E_vdw_ ^c^ (kcal/mol)	E_elec_ ^d^ (kcal/mol)
1	−5.88	−7.08	−2.7	−0.25
2	−5.40	−6.59	−3.09	−0.28
3	−5.01	−6.20	−2.71	−0.19
4	−4.83	−6.03	−2.92	−0.22
5	−4.61	−5.80	−2.45	−0.12
6	−4.49	−5.68	−2.27	−0.26
7	−4.49	−5.68	−2.13	−0.18
8	−4.41	−5.61	−2.88	−0.24
9	−4.40	−5.59	−2.03	−0.18
10	−4.40	−5.60	−2.58	−0.10

^a^ Binding energy. ^b^ Intermolecular energy. ^c^ Van der Waals energies. ^d^ Electrostatic interactions.

## Data Availability

Not applicable.

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
