# Peer review of "Spiro-Oxindole Skeleton Compounds Are Efficient Inhibitors for Indoleamine 2,3-Dioxygenase 1: An Attractive Target for Tumor Immunotherapy"

_ijms, 2022, doi:10.3390/ijms23094668_

Round 1
Reviewer 1 Report
The authors have addressed previous comments and the manuscript is now suitable for publication. There are few things to note;
- Table 3: What is the unit for all the energies reported? kcal/mol?? Add this information.
- Table S1, S2 and S3, add kcal/mol
- Need to read the manuscript to improve the use of language and grammar
- Noticed that for the synthesized compounds high resolution mass spectral data is not given. This should be given for new compounds only
Author Response
Comment 1: Table 3: What is the unit for all the energies reported? kcal/mol?? Add this information.
Response:Thanks for your nice suggestion. The unit for all the energies reported was added in revised manuscript.
Comment 2: Table S1, S2 and S3, add kcal/mol
Response:Thanks for your suggestion. The unit for all the energies reported was added in revised SI file.
Comment 3: Need to read the manuscript to improve the use of language and grammar
Response:Thanks. Language and grammar have been improved in revised manuscript.
Comment 4: Noticed that for the synthesized compounds high resolution mass spectral data is not given. This should be given for new compounds only.
Response:Thanks for your nice suggestion. Mass spectrometry was from laboratory instrument sharing platform in our department, we just received the image of full mass spectrum to confirm molecular weight of compounds. Due to our negligence, we did not copy the original data. We are so sorry that the high-resolution mass spectral data is not available in the SI file.
Reviewer 2 Report
The authors have responded to all of my comments, and made appropriate corrections in the text of the present version manuscript. Thus, I would like to recommend the publication of the re-revised manuscript in International Journal of Molecular Sciences in present form.
Author Response
Comment: The authors have responded to all of my comments, and made appropriate corrections in the text of the present version manuscript. Thus, I would like to recommend the publication of the re-revised manuscript in International Journal of Molecular Sciences in present form.
Response:Thank you very much. It is a great honor to have your recognition of our work.